# Peritumor Mucosa in Advanced Laryngeal Carcinoma Exhibits an Aberrant Proangiogenic Signature Distinctive from the Expression Pattern in Adjacent Tumor Tissue

**DOI:** 10.3390/cells13070633

**Published:** 2024-04-05

**Authors:** Silva G. Kyurkchiyan, Gergana Stancheva, Veronika Petkova, Stiliana Panova, Venera Dobriyanova, Iglika Stancheva, Venelin Marinov, Zahari Zahariev, Radka P. Kaneva, Todor M. Popov

**Affiliations:** 1Molecular Medicine Center, Medical University—Sofia, 1463 Sofia, Bulgaria; sggiragosyan@abv.bg (S.G.K.); rkaneva@medfac.mu-sofia.bg (R.P.K.); 2Department of ENT, Medical University—Sofia, 1463 Sofia, Bulgaria

**Keywords:** field cancerization, laryngeal carcinoma, HIF, HIF switch, HIF-1α, HIF-2α, HIF-3α, VEGF-A, ETS-1, miR-210

## Abstract

The field cancerization theory is an important paradigm in head and neck carcinoma as its oncological repercussions affect treatment outcomes in diverse ways. The aim of this study is to assess the possible interconnection between peritumor mucosa and the process of tumor neoangiogenesis. Sixty patients with advanced laryngeal carcinoma were enrolled in this study. The majority of patients express a canonical HIF-upregulated proangiogenic signature with almost complete predominancy of HIF-1α overexpression and normal expression levels of the HIF-2α isoform. Remarkably, more than 60% of the whole cohort also exhibited an HIF-upregulated proangiogenic signature in the peritumoral benign mucosa. Additionally, the latter subgroup had a distinctly shifted phenotype towards HIF-2α upregulation compared to the one in tumor tissue, i.e., a tendency towards an HIF switch is observed in contrast to the dominated by HIF-1α tumor phenotype. ETS-1 displays stable and identical significant overexpression in both the proangiogenic phenotypes present in tumor and peritumoral mucosa. In the current study, we report for the first time the existence of an abnormal proangiogenic expression profile present in the peritumoral mucosa in advanced laryngeal carcinoma when compared to paired distant laryngeal mucosa. Moreover, we describe a specific phenotype of this proangiogenic signature that is significantly different from the one present in tumor tissue as we delineate both phenotypes, quantitively and qualitatively. This finding is cancer heterogeneity, per se, which extends beyond the “classical” borders of the malignancy, and it is proof of a strong interconnection between field cancerization and one of the classical hallmarks of cancer—the process of tumor neoangiogenesis.

## 1. Introduction

Laryngeal carcinoma is a worldwide burden and, despite advances in the individualization of treatment for a number of other solitary malignancies, curative treatment options in this subdomain of the oncology remain limited to surgery and/or chemoradiation therapy based on anatomic location and TNM staging [1,2]. One of the main theories for oncogenesis and recurrences in head and neck cancer is the field cancerization theory proposed by Slaughter [3]. According to it, on a molecular level, some processes of cell dysregulation characteristic of the malignancy are evident in clonal patches of the surrounding histologically normal mucosa around the tumor and, presumably, both have a common predecessor. Our team has already reported the existence of an aberrant microRNA signature in the peritumoral tissue surrounding an advanced laryngeal carcinoma that could be used as a biomarking tool for demarcating field cancerization [4]. Similar patterns of abnormal expressions of regulatory molecules in both tumor tissue and peritumoral mucosa are evidence of a common clonal heritage and simultaneously signify the existence of genomic instability in histologically normal peritumoral mucosa, which could determine a higher stochastic chance of carcinogenesis in the area of primary tumor resection later on. Along with that, angiogenesis is one of the classical hallmarks of cancer, and a vast number of studies have focused on investigating its characteristics in laryngeal carcinoma. Hypoxia-inducible factors and their induction of vascular endothelial factor-A form the main axis of the canonical proangiogenic pathway. Classically, hypoxia-inducible factor-1-aplha (HIF1-α) is most active during short periods (2–24 h) of intense hypoxia or anoxia (<0.1% O_2_), whereas HIF-2α could be active for a longer period of mild hypoxia (<5% O_2_). This phenomenon is described in the literature as the HIF ‘switch’—HIF-1α drives the initial response to hypoxia, but afterwards HIF-2 overtakes the major role during chronic hypoxic exposure. Despite the fact that most malignancies display the characteristics of a HIF switch, our team has already shown the predominating role of HIF1-α in laryngeal carcinoma, e.g., the lack of such an HIF switch [5]. 

In the light of those findings, the aim of our study was to evaluate the possible existence of an abnormal proangiogenic expression profile in histologically normal peritumoral mucosa and, if evident, compare the expression pattern with that seen in tumor samples. In the current study, for the first time in the literature we report the existence of an abnormal proangiogenic expression profile present in peritumoral mucosa in advanced laryngeal carcinoma when compared to paired distant laryngeal mucosa. Moreover, we describe a specific phenotype of this proangiogenic signature that is significantly different from the one present in tumor tissue as we delineate both phenotypes, quantitively and qualitatively. 

## 2. Materials and Methods

### 2.1. Study Design and Treatment Protocols

A total of sixty patients (mean age at diagnosis 64.6 ± 8.7 years) with pathologically verified primary laryngeal carcinoma were recruited into the current prospective prognostic study. Their diagnoses were made based on current standards. The patients were admitted to the Department of ENT, Head & Neck Surgery, Medical University, Sofia, during the years 2018–2019, during which they underwent primary laryngectomy. During the surgery, four samples from each patient were obtained: tumor site—surface, tumor site—depth, histologically healthy peritumor mucosa within 1 cm from the border of the tumor, and paired normal laryngeal mucosa distant to the tumor (contralateral, at least 3 cm distance). All samples were stored in RNAlater Solutions (Thermo Fisher Scientific, Waltham, MA, USA) and frozen at −20 °C for a short period until they were transported to the Molecular Medicine Center, Department of Medical Chemistry and Biochemistry, Medical University, Sofia, and maintained at −80 °C until use.

All the patients signed an individual written consent form. The study was approved by the Ethics Committee of Medical University, Sofia (KENIMUS, approval code: BK-373 from 11.04.2016 and approval code: BK-329 from 08.03.2021). The enrolled cohort was a single-surgeon consecutive series, and the inclusion criteria have been described in a previous study [4]. All the enrolled patients were histologically diagnosed with advanced-stage squamous cell carcinoma of the larynx (T3/T4 stage), and all of them underwent primary laryngectomy with free resection margins and neck dissection ipsilaterally (2–5 levels). In the cases of tumors crossing the median line, contralateral neck dissection was also performed (2–5 levels). Additionally, if the tumor extended into the subglottic or retrocricoid region, a full paratracheal lymph node dissection (levels 6–7) was carried out. All the patients underwent postoperative radiotherapy or combined chemoradiotherapy according to the protocol. The follow-up period was an average of 24 months, with a standard deviation of 13 months. The patients were followed-up every month during the first six months after surgery and every three months after this period.

### 2.2. RNA Extraction

Total RNA (including miRNAs) was isolated from 60 fresh-frozen tumor materials and adjacent normal tissue by using the miRNeasy Mini Kit (Qiagen, Hilden, Germany) according to the manufacturer’s recommendations. The RNA concentration was determined with a NanoDrop 2000 Spectrophotometer (Thermo Fisher Scientific, MA, USA). The Qubit™ RNA HS Assay Kit, Qubit™ 2.0 Fluorometer (Life Technologies, Thermo Fisher Scientific Inc.) was used for the precise quantification of 50 ng/µL and 100 ng/µL RNA dilutions, which were further used for miRNA array and RT-qPCR analysis, respectively. The RNA integrity numbers (RIN) were measured by using the 2100 Bioanalyzer (Agilent, Santa Clara, CA, USA) as described in a previous study [4].

### 2.3. Real-Time Quantitative Polymerase Chain Reaction (RT-qPCR)

Quantitative real-time PCR was performed in a 7900HT Fast Real-Time PCR System machine (Applied Biosystems, Foster City, CA, USA) with primers for the selected mature RNAs, HIF-1α, HIF-2α, HIF-3α, VEGF-A, VEGFR1, VEGFR2, ETS-1, and miR-210, as QuantiTect Primer Assay and miScript Primer assays, respectively, designed by Qiagen (Qiagen, Hilden, Germany). The reactions were performed in triplicate with a total volume of 10 μL, and the mixtures for the mRNAs and miRNAs included respective SYBR Green PCR Mix (Qiagen, Hilden, Germany), Primer Assay (Qiagen, Hilden, Germany) for the respective genes, and miRNA and cDNA synthesized from 400 ng of total RNA using the miScript II RT Kit (Qiagen, Hilden, Germany) according to the manufacturer’s recommendations. Negative and no-template controls were also evaluated. The relative quantifications (RQs) of mRNAs as well as miRNAs in samples were analyzed by the 2^−ΔΔCt^ method, which has been commonly used for relative quantification in RT-qPCR data analysis. This method is a convenient way to calculate relative changes in gene expression levels between different samples by directly using the threshold cycles (CTs) generated from real-time quantitative PCR experiments [6].

Sample data were normalized to β-actin for all the mRNAs and to the U6 snRNA level for miR210 (as an internal control). ∆Ct represents the difference between the target gene and the assay’s reference gene, often called a housekeeping gene or internal control, to control for variance among samples using following formula: ∆Ct = Ct (gene/miRNA of interest) − Ct (housekeeping gene/internal controls). Meanwhile, the ∆∆Ct for each sample was calculated by comparing the calculated ΔCT values to those generated by the results from the normal samples, which were considered to be a baseline, using following formula: ∆∆Ct = ∆Ct (Tumor sample) − ∆Ct (adjacent normal sample). The fold change (RQ) was calculated as 2^−ΔΔCT^, RQ values over 2.00 were defined as showing overexpression, and RQ values less than 0.5 were defined as showing underexpression.

### 2.4. Statistical Analysis

Data analysis was performed with SPSS software ver. 23.0 for Windows (IBM SPSS, Armonk, NY, USA) and GraphPad Prism software 9.0. The Kolmogorov–Smirnov test for normality, Wilcoxon test, Mann–Whitney U test, Kruskal–Wallis or one-way ANOVA test, and paired and unpaired *t*-tests were used as appropriate. The relations between two continuous variables were evaluated using the bivariate correlation coefficient Spearman’s rank test. Correlations between the expression levels of miRNAs and clinicopathological features were analyzed by using Kruskal–Wallis rank tests for k independent samples and the Mann–Whitney U test for independent association analysis between any two subgroups. The Friedman test was used to determine differences between the tumor, peritumor mucosa, and control laryngeal mucosa. Recurrence-free survival was calculated from the date of diagnosis until recurrence or death caused by the malignancy was registered. Survival curves were plotted through the Kaplan–Meier method, and the log-rank test was used to compare survival between groups. Additionally, univariate and multivariate Cox regression analyses were performed to test the associations of various factors with survival time. A two-tailed *p*-value ≤ 0.05 was considered significant.

## 3. Results

The study group included predominantly HPV-negative tumors (90.3%), which were validated with p16 immunohistochemistry, with all patients having a history of long-term smoking. In terms of tumor staging, the majority of the cases (87.1%) were classified as pT4a, one case was pT4b, and the rest were staged as pT3. Almost half of the group, 48.3% of the cases, had pathologically verified metastatic processes, and the distribution of N status was as follows: N1 (28.6%); N2a/N2b/N2c (57.1%), N3 (14.3%). All the clinicopathological characteristics of the whole cohort have been summarized in Table 1.

The majority of patients expressed a canonical HIF-upregulated proangiogenic signature with almost complete predominancy of mRNA HIF-1α overexpression and normal expression levels of the mRNA HIF-2α isoform, i.e., HIF switch was not present in advanced laryngeal cancer.

More than half of patients (56%) with advanced laryngeal carcinoma presented with mRNA HIF-1α upregulation (RQ > 2), and mRNA HIF-2α overexpression was evident in only 10% of the cases [Figure 1]. Additionally, when a direct comparison of the RQ expression levels of both isoforms was performed, HIF-1α had unambiguously higher expression levels in comparison to its second isoform in 93.3% of all tumor samples. These data were obtained with the Wilcoxon ranking test, which undeniably showed significantly higher expression levels of mRNA HIF-1α vs. mRNA HIF-2α (*p* < 0.001, z = 6.35). In summary, we observed a canonical proangiogenic phenotype in advanced laryngeal cancer with a significant upregulation of HIF-1α but not HIF-2α, i.e., HIF switch was not evident.

Remarkably, the majority of patients (60%) also exhibited an HIF-upregulated proangiogenic signature in the peritumoral benign mucosa. Additionally, the peritumoral mucosa displayed a distinctly shifted phenotype towards HIF-2α upregulation compared to the “no-HIF switch” phenotype in the tumor tissue, i.e., a tendency towards an HIF switch was observed in the peritumoral mucosa samples.

The analysis of the samples from the histologically normal mucosa adjacent to the tumor tissue revealed that in 60% of all cases, there was an aberrant proangiogenic overexpression of either, or both, HIF-1α/HIF-2α when compared to the paired distant laryngeal mucosa along with other classical molecules from the canonical proangiogenic cascade. Moreover, when analyzed, the expression patterns of those molecules differentiated significantly from the pattern evident in the paired tumor samples. First, despite still showing an upregulation of HIF-1α in 40% of cases (versus 56% in tumor samples), the RQ expression levels were significantly lower in the peritumoral tissue when compared to the tumor tissue (Wilcoxon test, *p* = 0.03, z = 2.16) [Figure 1]. Even more pronounced was the change in HIF-2α expression—we detected significantly higher levels of HIF-2α expression in the peritumoral mucosa when compared to the paired tumor samples (Wilcoxon test, *p* < 0.001, z = 5.44). Thus, we saw two distinct abnormal proangiogenic signatures in the tumor and the paired adjacent normal peritumoral mucosa—we recognize the leading role of the overexpressed HIF-1α in the tumor samples, whereas there is an evident shift towards HIF-2α upregulation and lowering of the expression of the first isoform in the peritumor mucosa. Ergo, there is a shift towards a “HIF switch” in the peritumoral mucosa in contrast to the tumor tissue, where HIF-1α holds the predominant role [Figure 2]. The figure was adapted from Serocki et al. [7], 2018, after acquiring permission and was primarily validated on endothelial cells. Despite this, both the distinctive phenotypes described in our study on tumor/peritumor samples correspond very well to this model. Additionally, the other major molecules from this cascade had significantly higher expression in the peritumoral mucosa when compared to the paired distant control laryngeal mucosa: VEGF-A overexpression was displayed in 40% of patients, ETS-1 in 42%, VEGFR1 in 36.7%, and VEGFR2 in 41.7%. Interestingly, the VEGF-A expression levels in the peritumor tissue were significantly lower than those in the tumor tissue (*p* < 0.001, z = 3.931), while the VEGFR2 levels were significantly higher in the peritumoral mucosa in comparison to the tumor tissue (*p* < 0.001, z = 4.56) [Figure 1].

The HIF-3α expression levels were significantly higher in the peritumoral mucosa as part of the HIF switch, but no definite association can be established with the other proangiogenic molecules.

HIF-3α is a molecule that was massively found to be silenced in 78.3% of the tumor samples (RQ < 0.5). In the peritumoral mucosa samples, we saw a statistically significant rise in the expression levels of HIF-3α when compared to the paired tumor samples (Wilcoxon test, *p* = 0.006, z = 2.75) despite this still largely downregulated (55%) in relation to the paired control samples (RQ < 0.5) [Figure 1]. When we analyze correlations with the other molecules in both the tumor and the peritumoral mucosa, we obtain *p*-values beneath 0.05 (Pearson’s correlation); however, visual evaluation of the scatterplot does not validate these results since the relationship is not monotonic [Figure 3, last column].

Mir-210 is overexpressed in the tumor tissue but does not show a well-established association with any of the main proangiogenic genes, including HIF-1α.

Among the listed differences in the proangiogenic expression patterns between the tumor and the peritumoral mucosa, the expression of angiomir miR-210 also aligns. In the tumor tissues, we recorded that almost half of the patients displayed an overexpression of miR-210 (48.3%), which was in contrast to the 16.6% overexpression rate of this molecule in the peritumoral mucosa. Additionally, pairwise comparison with the Wilcoxon rank test revealed significantly higher levels of expression of miR-210 in the tumor tissue compared to the peritumoral mucosa (*p* < 0.001, z = 5.03) [Figure 1]. Contrary to expectations and the literature data, when correlation tests were run between miR-210 and HIF-1α, HIF-2α, VEGF-A, VEGFR1, and VEGFR2, visual evaluation of the scatterplots concluded that there was no clear monotonic relationship between the expression levels of this microRNA and the other proangiogenic molecules, despite our obtaining certain p-values beneath 0.05 [Figure 3, first row].

ETS-1 displayed stable and identical significant overexpression in both the proangiogenic phenotypes present in the tumor and the peritumoral mucosa. The correlations between the proangiogenic molecules in both sample groups match, but they tend to differ in terms of strength.

The ETS-1 molecule was overexpressed to a similar extent in both the tumor and the peritumoral mucosa (45% vs. 42%), and no statistically significant difference was reported. The investigated classical proangiogenic molecules in the tumor tissue displayed strong correlations with each other, which can be seen in detail in the scatter dot matrix in Figure 3. We additionally analyzed the correlations of those molecules with their expression profiles in the peritumor tissue, and they did not differ widely from those in the tumor samples in terms of significance. Nevertheless, when comparing the strength of association, one could undoubtedly distinguish the difference. To compare the strength of association among these molecules, we created two heatmaps with the correlation coefficients (Spearman’s ρ (rho) or *r_s_*), as given in Figure 4. Firstly, we identified that in contrast to the tumor tissue, where we found a moderate correlation between HIF-1α and HIF-2α, but such an association was not valid in the peritumoral tissue. ETS-1 had a far stronger association with VEGF-A in the peritumoral tissue compared to the tumor tissue (*r_s_* = 0.32 vs. 0.63). HIF-1α/HIF-2α correlated to a far greater extent with the two VEGF receptors and ETS-1 in the tumor tissue compared to the peritumor tissue, their level of association with VEGF-A was identical in both sample groups. Additionally, the association between VEGF-A and VEGFR1 displayed a higher correlation coefficient than VEGF-A and VEGFR2 in the tumor tissue, and this relation was inverted in the peritumoral mucosa [Figure 4].

## 4. Discussion

Several studies have investigated and displayed, on a cellular level, the molecular dysregulation of histologically “normal” peritumoral tissue that bears identical genetic and epigenetic traits to its adjacent malignancy due to their probable common monoclonal origin [8]. The current study is to our knowledge the first to address the possibility that peritumoral mucosa could express an aberrant proangiogenic signal. The results from the mRNA expression analysis of the main proangiogenic genes undoubtedly reveal the existence of such a pathological activity [Figure 1].

It is a well-known fact that in normoxia HIFs are degraded via the pVHL-E3 ubiquitin ligase complex, whereas in a hypoxic environment HIF-1α/HIF-2α stabilize and translocate to the nucleus where they induce their effect by binding to hypoxia-responsive elements (HREs) [9]. As peritumoral mucosa by an axiom has a normal supply of oxygen, we can hypothesize two possible explanations for these findings—either there is some unrecognized hypoxia in the peritumor microenvironment, or we can observe the intracellular processes of normoxic upregulation of the HIF molecules. The first alternative could be rationalized with the existence of a “steal phenomenon”, whereby tumor blood flow could decrease peritumoral perfusion locally, leading to a certain level of hypoxic microenvironment surrounding the tumor. The second alternative involves assuming the existence of an intracellular non-canonical proangiogenic phenotype. It has been observed that aggressive cancers constitutively express hypoxia-related proteins (HRPs) even in the presence of oxygen, a condition known as pseudohypoxia [10,11]. Such a phenotype is closely related to the so-called “acid-mediated invasion hypothesis”, which states that the expression of glycolytic proteins at the edge of tumors creates an acidic microenvironment, which could induce apoptosis in stromal cells, resulting in space being created which tumor cells can proliferate into [12]. We could speculate that such changes in the peritumor environment could lead to phenotype changes in the surrounding tissues and result in aberrant phenotypes. Such a hypothesis is supported by the fact that we see a strong correlation between the expression levels of MMP2 in the tumor and the peritumor mucosa (*p* < 0.001, z = 0.528, unpublished data), and acid has been shown to induce the release of matrix metalloproteinases, resulting in degradation of the matrix [10]. Moreover, a number of studies have reported on pseudohypoxic tumor phenotypes that occur due to specific mutations that upregulate HIF-1α and prevent its degradation, e.g., mutations in isocitrate dehydrogenase (IDH) [11], (PTEN), von Hippel–Lindau protein (pVHL), p53, epidermal growth factor (EGF), and mutant Ras and Src [13,14]. In light of the field cancerization theory, we could speculate that a possible explanation for this aberrant proangiogenic activity in peritumoral mucosa could be the presence of some mutations that are found in both tumor and peritumoral cells due to their common monoclonal ancestry. Future studies in this subfield of oncological research should find the answers to all of these possible explanations.

Another important aspect of the findings of our study is the difference in the proangiogenic phenotypes between the tumor tissue and the peritumoral mucosa. Due to the inversion of the HIF-1α/HIF-2α expression ratios in the peritumor samples, we could categorize this phenotype as being more shifted towards an “HIF switch” in contrast to the tumor phenotype dominated by HIF-1α. Firstly, our data once again confirm a finding from a previous study of our team, which excluded the existence of such an HIF switch in advanced laryngeal carcinoma, a phenotype that is typical in many other malignancies and that is characteristic of advanced cancer [5]. This difference in phenotypes is illustrated in Figure 2, which is an adapted graphic from the magnificent review on the HIF switch regulation published by the Rafał Bartoszewski’s team (with permission of the authors) [7]. This variance in the expression levels is cancer heterogeneity, per se, which extends beyond the “classical” borders of the malignancy. As a consequence, this would inevitably affect tumor resistance in terms of treatment. Beneath the shift in the HIF isoform’s expression levels, one can also identify a rearrangement of the association’s strength between the major proangiogenic genes [Figure 4], which could be a consequence of a change in the regulatory framework. An important fact that should be considered is that the stability of the expression levels of ETS-1 does not change significantly in contrast to the other major proangiogenic molecules. These findings could position ETS-1 as a more independent gene in terms of its regulation of the canonical cascade of neoangiogenesis.

A shortcoming of the current study and simultaneously a future perspective is the need to visualize the described differences in expression levels of the studied proangiogenic molecules. In light of these perspectives, the methodological limitations of immunohistochemistry in terms of sensitivity when compared to RT-PCR are noteworthy [15].

## 5. Conclusions

To our knowledge, this is the first comprehensive study to uncover the existence of a pathophysiological proangiogenic signature in the peritumor mucosa in head and neck cancer. Additionally, the patterns of expression of the major proangiogenic molecules are distinctly different in comparison to the tumor proangiogenic signal. This finding is cancer heterogeneity, per se, which extends beyond the “classical” borders of the malignancy and is proof of a strong interconnection between field cancerization and one of the classical hallmarks of cancer—the process of tumor neoangiogenesis.

## Figures and Tables

**Figure 1 cells-13-00633-f001:**
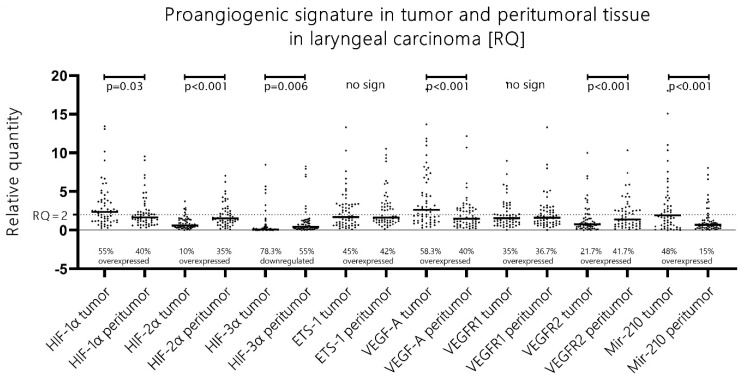
Major proangiogenic molecule expression profiles in tumor and peritumor mucosa samples with pairwise comparison; Wilcoxon test. Relative quantity expression levels of each of the genes are visualized with bee swarm boxplots, and tumor and peritumoral samples are compared.

**Figure 2 cells-13-00633-f002:**
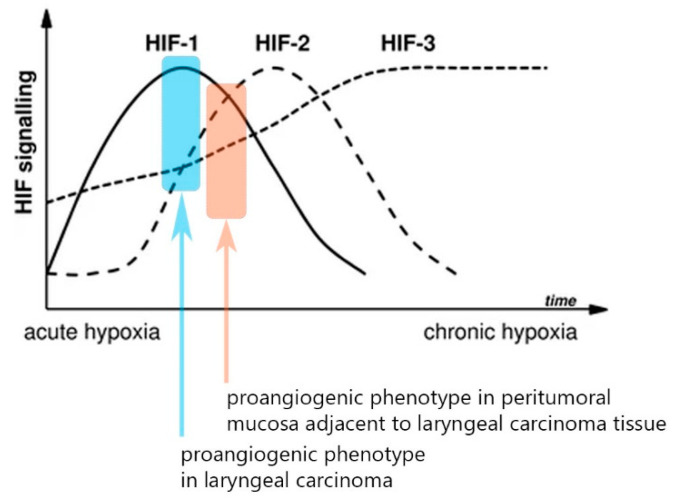
Schematic representation of HIF switch regulation with the two distinct signatures from the current study superimposed. Adapted from Serocki et al. [7] with permission.

**Figure 3 cells-13-00633-f003:**
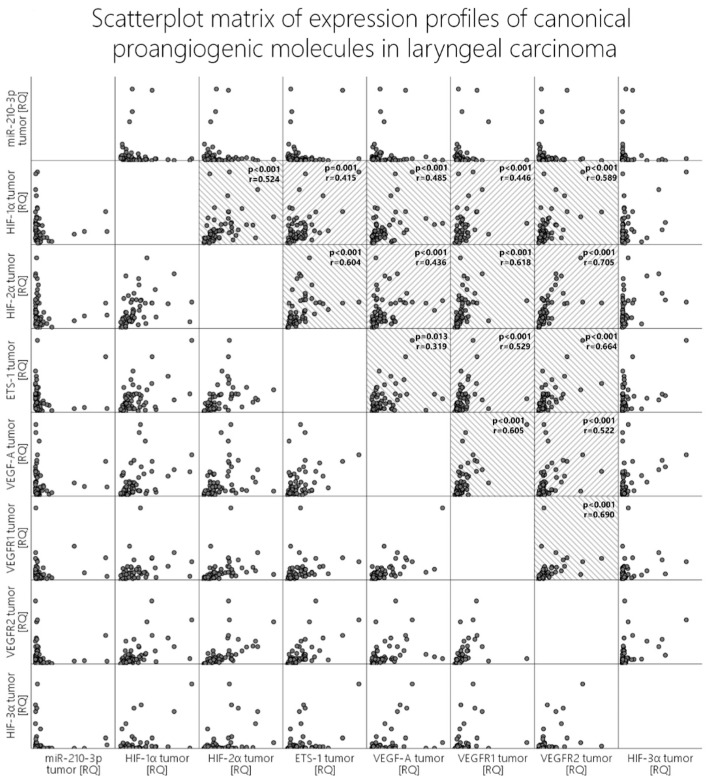
Scatterplot matrix of expression profiles of canonical proangiogenic molecules in laryngeal carcinoma. Statistically significant correlations are marked with gray semi-transparent interlace stripes. P-values and correlation coefficients are given in each scatterplot box.

**Figure 4 cells-13-00633-f004:**
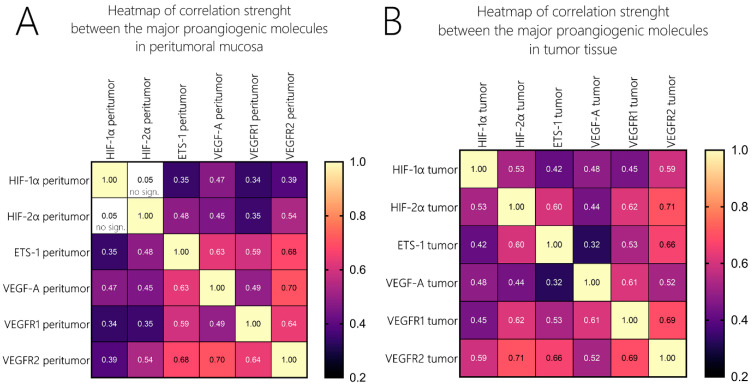
Heatmaps of correlation coefficients (ρ) in tumor and peritumor samples. Strength of correlation is visualized as a heatmap—lighter red/yellow marks strong correlation patterns and dark magenta/purple marks weaker correlation strength. Correlation coefficients (ρ) are given for each correlation. (**A**) heatmap of expression correlation strength in peritumor mucosa, (**B**): heatmap of expression correlation strength in tumor tissue.

**Table 1 cells-13-00633-t001:** Clinicopathological characteristics of the patients enrolled in this study.

Characteristics	Number of Patients	Percentage
**Age**	64 (mean), 46–83	N/A
**Gender**	58 males, 2 females	96.6% vs. 3.3%
**T-stage**		
pT3	7	11.6%
pT4a & pT4b	53	88.3%
**N-stage**		
pN0	32	53.3%
pN1	7	11.6%
pN2a	2	3.3%
pN2b	8	13.3%
pN2c	7	11.6%
pN3	4	6.6%
**Metastasis (pN0 vs. pN1–3)**		
pN0	32	53.3%
pN1–3	28	46.7%
**Grade**		
G1	15	25%
G2	42	70%
3	3	5%

## Data Availability

Raw data were generated at Medical University, Sofia. The derived data supporting the findings of this study are available from the corresponding author [T.M.P.] upon request.

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
