# Peer review of "Peritumor Mucosa in Advanced Laryngeal Carcinoma Exhibits an Aberrant Proangiogenic Signature Distinctive from the Expression Pattern in Adjacent Tumor Tissue"

_cells, 2024, doi:10.3390/cells13070633_

Round 1

Reviewer 1 Report

Comments and Suggestions for Authors

Summary:

The authors analyzed and compared the mRNA expression level of HIF1α, HIF2 α, ETS, VEGF-A, VEGFR1 and VEGFR2, and MirRNA -210 in tumor and peritumor areas in sixty patients with advanced laryngeal carcinoma. The results demonstrate that the majority of patients express a HIF1α mRNA upregulation and normal mRNA level of HIF2α and HIF-switch is not present in advanced laryngeal tumor, however in the peritumor mucosa the HIF-switch is observed. ETS-1 shows  significantly overexpression in both proangiogenic phenotypes present in tumor and peritumoral mucosa. In the study, the authors described that they report for the first time the existence of an abnormal proangiogenic expression profile present in the peritumoral mucosa in advanced laryngeal carcinoma.

 Suggestions:

11.  Add a table to list the clinicopathological characteristics of patients,         including age, gender, histological type, stage, regional lymph node (N), and grade. Compare the HIF1a overexpression and non-overexpression groups to check any difference in the clinicopathological characteristics.

22.  Include IHC staining for the specimen: check the protein expression level of HIF1α and HIF2αrelated to the clinicopathological characteristics of patients.

33. In lines 103-122: missing mRNA when mentioning HIF1α and HIF2 α levels.

44. In line 123: Figure 1 is missing the p values of ETS1 and VEGFR1(tumor compared to peritumor). Moreover, the gene names can list above the p-value or below the tumor and peritumor. Otherwise, it will cause confuse to the reader.

55. In lines 127-128: The later and the one causing confusion can directly use the whole name on them.

66. In line 153, Figure 2 is adapted from Serocki et al., 2018 which validated in endothelial cells if this phenomenon can be used in tumor models is still questionable. Furthermore, some of the miRNAs in the figure are not directly related to this study.

77. In line 252, Figure 3 should be corrected to Figure 2.

Comments on the Quality of English Language

Moderate editing of English language required

Author Response

Dear Reviewer,

Thank you for your careful consideration regarding our manuscript submission ID cells-2921841 with title "Peritumor mucosa in advanced laryngeal carcinoma exhibits an aberrant proangiogenic signature distinctive from the expression pattern in adjacent tumor tissue ".

Following the helpful comments, we revised our work. The general lines of the revision of the manuscript were carried out in accordance with the points of criticism. 

Please check attachment for the fully revised manuscript.

Reviewer #1 comments:

Regarding:

“11.  Add a table to list the clinicopathological characteristics of patients,         including age, gender, histological type, stage, regional lymph node (N), and grade. Compare the HIF1a overexpression and non-overexpression groups to check any difference in the clinicopathological characteristics.”

Answer to 1st suggestion: We have added a table with the clinicopathological characteristics of the cohort with reference to it.  We have compared HIF1A expression levels with the clinicopathological characteristics but we could not find any statistically significant associations, respectively we did not specifically highlight that in the text.

Regarding:

“22.  Include IHC staining for the specimen: check the protein expression level of HIF1α and HIF2αrelated to the clinicopathological characteristics of patients.”

Answer to 2nd suggestion: We are planning to try to visualize the differences with IHC but it would not be possible to make this in such short term in order to include it in this paper. We added this as shortcoming of the article and future perspectives (see end of discussion in revised manuscript). In the light of these perspectives worthy of notice is also the methodological limitations of immunohistochemistry in terms of sensitivity when compared to RT-PCR and we should acknowledge also that possibility.

Regarding:

  1. In lines 103-122: missing mRNA when mentioning HIF1α and HIF2 α levels.

Answer to 3rd suggestion: “mRNA” clarifications have been added to the cited paragraph.

Regarding:

  1. In line 123: Figure 1 is missing the p values of ETS1 and VEGFR1(tumor compared to peritumor). Moreover, the gene names can list above the p-value or below the tumor and peritumor. Otherwise, it will cause confuse to the reader.

Answer to 4th suggestion: Since there is no statistical significance, we omitted those p-values on purpose, but now we have edited the figure and added “no sign” label for clarity. About the second part of the suggestions - we understand that gene names with tumor/peritumor labelling are visually overloaded, but the graphic is automatically generated by the GraphPad Prism software and we have just manually added the p-values and the overexpression percentages.

Regarding:

  1. In lines 127-128: The later and the one causing confusion can directly use the whole name on them.

Answer to 5th suggestion: Paragraph has been revised as suggested.

Regarding:

  1. In line 153, Figure 2 is adapted from Serocki et al., 2018 which validated in endothelial cells if this phenomenon can be used in tumor models is still questionable. Furthermore, some of the miRNAs in the figure are not directly related to this study.

Answer to 6th suggestion: We fully understand that Serocki’s figure displays data validated in endothelial cells but still the described distinctive phenotypes in our study in tumor/peritumor samples correspond very well to this model and we would like to keep this reference. We added notification of your suggestion in the text: “The figure is adapted from Serocki et al, 2018 after acquiring permission and primarily is validated on endothelial cells. Despite this, both distinctive phenotypes described in our study in tumor/peritumor samples correspond very well to this model.” Additionally, we have removed the microRNA labels from the figure as suggested.

Regarding:

  1. In line 252, Figure 3 should be corrected to Figure 2.

Answer to 7th suggestion: Numbering has been revised as suggested.

I hope that the revised version of the manuscript will be satisfying and you would consider it suitable for publication in “Cells”.

With kind regards

Reviewer 2 Report

Comments and Suggestions for Authors

the manuscript entitled "Peritumor Mucosa in Advanced Laryngeal Carcinoma Exhibits 2
an Aberrant Proangiogenic Signature Distinctive from the 3
Expression Pattern in Adjacent Tumor Tissue" is well written.

From a technical point of view it is not clear how the 2-ddct methods was performed. for example figure 1 say pairwise comparison what is the calibrator/reference sample in each case ?

From a biological point of view how is this signature correlated to smoking, and how is it correlated to the secreted angiogenic factors from the tumor

Comments on the Quality of English Language

the manuscript is well written, can be a bit more descriptive

Author Response

Dear Reviewer,

Thank you for your careful consideration regarding our manuscript submission ID cells-2921841 with title "Peritumor mucosa in advanced laryngeal carcinoma exhibits an aberrant proangiogenic signature distinctive from the expression pattern in adjacent tumor tissue ".

Following the helpful comments, we revised our work. The general lines of the revision of the manuscript were carried out in accordance with the points of criticism.

Please see the fully revised manuscript as an attached file.

Reviewer #2 comments:

Regarding:

"From a technical point of view it is not clear how the 2-ddct methods was performed. for example figure 1 say pairwise comparison what is the calibrator/reference sample in each case ?"

Answer to 1st suggestion: From each patient we collected two samples from tumor tissue, one sample from histologically healthy peritumoral tissue and one control sample from contralateral distant mucosa, which was the reference control sample for the 2-ddct method. We have added detailed description of the method along with additional citation in the materials and methods section. 

Regarding:

"From a biological point of view how is this signature correlated to smoking, and how is it correlated to the secreted angiogenic factors from the tumor"

Answer to 2nd suggestion: There are numerous studies which show that smoking induces a significant number of those molecules. We have managed this bias with pairwise control samples taken from the same patient:  if smoking was one of the factors for overexpression, it would have induced similar expression levels in the contralateral control mucosa sample and thus we would had not detected significant overexpression when referenced with the 2-ddct method. Additionally, the majority of our patients are smokers, so in that direction the group is almost homogenous and no correlations or associations could be sought with other characteristics.

I hope that the revised version of the manuscript will be satisfying and you would consider it suitable for publication in “Cells”.

With kind regards

Round 2

Reviewer 1 Report

Comments and Suggestions for Authors

Thanks for the revision.

Suggestion:

All figures only have titles but lack figure legends to highlight the main results in the figures.

In line 360, the reference should be number 3 instead of 5. There are missing references 4 and 5.

Comments on the Quality of English Language

Minor editing of English language required.

Author Response

Dear Reviewer,

Thank you for your careful consideration regarding our manuscript submission ID cells-2921841 with title "Peritumor mucosa in advanced laryngeal carcinoma exhibits an aberrant proangiogenic signature distinctive from the expression pattern in adjacent tumor tissue ".

Following the helpful comments, we revised our work. The general lines of the revision of the manuscript were carried out in accordance with the points of criticism.

Reviewer #1 comments:

Regarding:

“All figures only have titles but lack figure legends to highlight the main results in the figures.”

Answer to 1st suggestion: We have added additional description beneath each figure for clarity as suggested.

Regarding:

“In line 360, the reference should be number 3 instead of 5. There are missing references 4 and 5.”

Answer to 2nd suggestion: Due to technical formatting mistake, references were not numbered properly. We have formatted the references in a proper manner.

I hope that the revised version of the manuscript will be satisfying and you would consider it suitable for publication in “Cells”.

Looking forward to your final decision,

Kind regards

Reviewer 2 Report

Comments and Suggestions for Authors

Thank you for your response, the addition of technical technical details is sufficient for the reader to understand how the comparison was performed.

Author Response

Dear reviewer,

Thank you for your review, your helpful suggestions and your for your careful consideration regarding our manuscript submission.

Kind regards